# Transcutaneous Auricular Vagus Nerve Stimulation Alleviates Headache Symptoms in Migraine Model Mice by the Locus Coeruleus/Noradrenergic System: An Experimental Study in a Mouse Model of Migraine

**DOI:** 10.3390/biomedicines14010096

**Published:** 2026-01-02

**Authors:** Xingke Song, Zijie Chen, Haohan Zhu, Peijing Rong, Jinling Zhang, Xue Pu, Junying Wang

**Affiliations:** 1Institute of Acupuncture and Moxibustion, China Academy of Chinese Medical Sciences, Beijing 100700, China; a646565365@163.com (X.S.); kittotsuen@163.com (Z.C.); z109001515@163.com (H.Z.); jinlingzhang407@sina.com (J.Z.); puxue0209@163.com (X.P.); 2Institute of Basic Research in Clinical Medicine, China Academy of Chinese Medical Sciences, Beijing 100700, China; drrongpj@163.com

**Keywords:** transcutaneous auricular vagus nerve stimulation, migraine, locus coeruleus, norepinephrine system, analgesic mechanism

## Abstract

**Background/Objectives**: Migraine is a complex neurological headache disorder, and transcutaneous auricular vagus nerve stimulation (taVNS) can effectively relieve headache symptoms, but its mechanism of effect is still unclear. This study aimed to explore the regulatory effects of taVNS on the locus coeruleus (LC) and the norepinephrine (NE) system in migraine mice. **Methods**: C57/BL6 mice were randomly assigned to four experimental groups: the control group, model group, taVNS group, and sham taVNS group. A migraine model was established by administration of nitroglycerin. Headache behaviors were assessed using the orofacial stimulation test (OST) and the mouse grimace scale (MGS). Immunofluorescence staining was conducted to evaluate the expression of NE neurons in the LC, while Western blotting was used to determine the expression levels of α-2A adrenergic receptors in the spinal trigeminal nucleus caudalis (Sp5C). Additionally, fiber-optic recording was employed to monitor the real-time dynamics of NE release in Sp5C. **Results**: After taVNS intervention, the drinking time of OST in the model mice was significantly prolonged(*p* < 0.05), and facial expression scores were reduced (*p* < 0.05). TaVNS increased the number of NE neurons in the LC (*p* < 0.05), promoted the release of NE in Sp5C (*p* < 0.05), and upregulated the expression of α-2A adrenergic receptors in Sp5C (*p* < 0.05). **Conclusions**: The analgesic effects of taVNS are related to the activation of the LC-NE system and the inhibition of the decrease in Sp5C in migraine mice.

## 1. Introduction

Migraine is a complex neurological disorder characterized by severe pulsatile headaches on one or both sides, often accompanied by nausea, vomiting, allergies to sound and light stimuli, and cognitive impairments, including deficits in executive function, memory, and attention. According to a disease burden study in 2016, migraine has become the second leading cause of disability [1]. These persistent symptoms significantly disrupt the daily lives of individuals, leading to a serious negative impact on the patient’s life. Migraine patients presented lower pressure pain thresholds values in all the assessed muscles and worse performances in cortical excitability and executive functions [2]. Epidemiological studies showed that the prevalence of migraine in the United States is 14.9%, and in Asia, it ranges from 8.4% to 12.7% [3]. Although pharmacological treatment (such as sodium valproate, topiramate, metoprolol, and propranolol) are recommended for migraine, they often lead to adverse reactions, including addiction, tolerance, and other long-term complications [4]. Due to these limitations, non-pharmacological and non-invasive therapeutic interventions have become a focus of research in recent years.

Auricular acupuncture therapy originated in ancient China and is a traditional Chinese medical treatment. Among migraine patients seeking non-pharmacological treatment, the use of auricular acupuncture therapy is quite common [5], and it is effective for migraine [6,7]. Our previous bibliometric analysis showed that the analgesic effect of auricular acupuncture in treating migraine might be produced by stimulating the auricular vagus nerve [8]. Some studies have shown that vagus nerve stimulation (VNS) can effectively alleviate clinical symptoms in migraine patients, demonstrating promising clinical prospects [9]. Transcutaneous auricular vagus nerve stimulation (taVNS), as a non-invasive neuroregulatory stimulation-based therapeutic intervention, avoids the surgical trauma of traditional VNS and promotes innovation and development of VNS. A meta-analysis suggested that taVNS could effectively reduce the pain intensity of chronic pain [10]. Repeated taVNS treatments significantly reduced the number of migraine attack days, headache pain intensity, and frequency of migraine attacks [11,12]. Another study also reported that taVNS significantly reduced headache duration for chronic migraine patients [13]. More and more evidence suggested that taVNS reduced the headache pain intensity and decreased the frequency of migraine attacks [14], but its mechanism is still unclear.

The theoretical basis of using taVNS to treat neurological diseases is to activate the central nervous system by stimulating the vagus nerve sensory fibers that are distributed in the periphery. The auricular branch of the vagus nerve, along with the remaining vagus nerve fibers, reaches the nucleus solitarius (NTS) of the brainstem, which has a prominent projection of the locus coeruleus (LC)–norepinephrine (NE) system [15]. In healthy human participants and migraine patients, acute taVNS can significantly regulate the pathways of the vagus nerve, including the NTS and LC [16]. Repeated taVNS also regulated the functional connectivity between the NTS and brain regions associated with the limbic system (bilateral hippocampus), as well as the functional connectivity between the LC and the brain regions that are closely related to pain processing and modulation (bilateral postcentral gyrus, thalamus) in migraine patients [17]. Another functional magnetic resonance imaging study also indicated that both 1 Hz and 20 Hz taVNS could modulate the functional connectivity between the LC and other brain regions in different ways [18]. The LC is a key brain region in the pathophysiological regulation of migraine and provides the main source of NE for the spinal cord and the dorsal horn of the neocortex. The reduction in NE signaling in the spinal cord inhibited the activation of the trigeminal nerve induced by the dura mater at the trigeminocervical complex level, which highlights the potential analgesic effect of LC inhibition on migraine-related pain [19]. Therefore, the activation of the vagus nerve conduction pathway, especially the regulation of the NTS-LC-NE pathway, may be one of the key links in the analgesic effect of taVNS on patients with migraine.

Thus, the regulatory effect of the LC-NE system may be involved in the analgesic effect of taVNS on migraine, but it is still unclear how taVNS exerts analgesic effects on migraine through the NE system. This study, firstly, demonstrated the analgesic effect of TAvns on migraine model mice. Secondly, this study aimed to clarify the analgesic mechanism of taVNS by investigating its effects on the expression of NE in the LC, its downstream projections to spinal trigeminal nucleus caudalis (Sp5C), and the regulation of the α-2A adrenergic receptor (α-2AAR) in Sp5C. Understanding these mechanisms will provide valuable insights for the analgesic effect of taVNS on migraine.

## 2. Materials and Methods

### 2.1. Animals and Grouping

Seventy SPF-grade, 8-week-old male C57BL/6J mice, with an average body weight of 20 g, were obtained from Beijing SPF Biotechnology Co., Ltd. (Production License Number: SCXK (Beijing) 2019–0010, Beijing, China). The animals were housed in the animal facility of the China Academy of Chinese Medical Sciences under controlled environmental conditions: temperature 20–25 °C, humidity (55 ± 5)%, and a 12 h light/dark cycle. They had free access to food and water.

The mice were randomly assigned to four experimental groups, the control group, model group, transcutaneous auricular vagus nerve stimulation (taVNS) group (auricular concha stimulation), and sham taVNS group (ear edge stimulation), with 16 mice in each group. For the fiber-optic recording experiment, the mice were further divided into two subgroups: the control + taVNS group and model + taVNS group, with a total of 6 mice in these groups.

All experimental procedures were approved by the Animal Ethics Committee of the Institute of Acupuncture and Moxibustion, China Academy of Chinese Medical Sciences (Approval No. D2022-03-28), and carried out in compliance with the Guide for the Care and Use of Laboratory Animals.

#### 2.1.1. Model Establishment

After a one-week acclimatization period in the animal facility, migraine modeling was initiated. Prior to modeling, nitroglycerin (NTG) (Beijing Yimin Pharmaceutical Co., Ltd., Beijing, China) stock solution (5 mg/mL) was diluted with saline to a working concentration of 1 mg/mL and stored in a sterile plastic centrifuge tube, protected from light. Fresh solutions were prepared for each use. C57BL/6J mice were weighed, and the corresponding NTG injection solution (10 mg/kg) was administered based on body weight [20]. Mice in the model group, taVNS group, and sham taVNS group underwent subcutaneous injection of NTG at the neck on days 1, 3, 5, 7, and 9 to establish a chronic migraine model. Mice in the model + taVNS group received a single intraperitoneal injection to establish an acute migraine model. If within 3–5 min, the mice exhibited signs such as reddening of both ears, frequent and rapid scratching of the head and face with their forepaws, climbing the cage frequently, and agitation, with a significant peak occurring between 30 and 90 min, it indicated successful modeling.

#### 2.1.2. TaVNS Intervention Method

Mice in the taVNS and sham taVNS groups underwent auricular concha stimulation and ear edge stimulation [21] on days 2, 4, 6, 8, and 10 of the modeling procedure, using the Hans-200A electroacupuncture device (Nanjing Jisheng Medical Technology Co., Ltd., Nanjing, China). Mice in the control + taVNS and model + taVNS groups received taVNS before modeling and 2 h after acute model induction. The specific stimulation parameters were as follows: sparse–dense waves, 2 Hz/15 Hz, intensity 1 mA, with chronic migraine treated for 30 min per session and acute migraine for 1 min per session (Figure 1A). The intervention was conducted daily between 9:00 a.m. and 16:00 p.m.

#### 2.1.3. The Orofacial Stimulation Test (OST)

In the orofacial stimulation experiment, the mice first acclimated to the room and the experimenter, followed by acclimation to the mechanical orofacial stimulation procedure. The mice made orofacial contact with the mechanical stimulation device (Ugo Basile, Gemonio, Italy), which consisted of metal wires connected to a mounting plate, in order to obtain a drinking reward (Figure 1B). The drinking duration was measured by detecting interruptions of an infrared light beam passing through the reward opening [22]. After each testing session, the bottom of the box was cleaned with 75% ethanol, and any remaining feces or residues were promptly removed. 

### 2.2. The Mouse Grimace Scale (MGS)

Mice were placed in a transparent plastic box (10 × 10 × 10 cm) for a 20 min acclimatization period. Subsequently, facial expressions of the mice were recorded for 2 min using the video recorder. The images exhibiting the greatest changes were selected, randomly arranged, and scored by three independent experimenters. The final score was determined by calculating the average of these ratings, with the scoring criteria being outlined in Table 1. All behavioral assessments were conducted in a controlled, quiet environment. After each testing session, the bottom of the box was cleaned with 75% ethanol, and any remaining feces or residues were promptly removed [23].

### 2.3. Immunofluorescence

Mice were perfused with saline, followed by a 4% paraformaldehyde solution. The brain tissues were then harvested and immersed in a 30% sucrose solution until the tissue had sunk to the bottom of the container. Each sample was sectioned into 40 μm thick slices (Fully automated cryostat, Leica, Nussloch, Germany).The locus coeruleus (LC) brain tissue slices of the mice were carefully dissected and blocked with a blocking solution containing 3% donkey serum, 0.5% Triton X-100, and 0.1 M phosphate-buffered saline (PBS). After incubating for 0.5 h, the blocking solution was removed. The slices were then incubated overnight at 4 °C with recombinant Anti-Tyrosine Hydroxylase antibody (1:200, ab137869, Abcam, Cambridge, UK). The next day, the tissue was washed three times with 0.1 M PBS for 5 min each. Following this, Alexa Fluor 488 donkey anti-rabbit IgG H + L (1:300, A21206, Thermo, Logan, UT, USA) was added, and the tissue was incubated at 24 °C in the dark for 1 h. The tissue was washed three times with 0.1 M PBS, mounted onto positively charged glass slides, and cover-slipped with 4′,6-diamidino-2-phenylindole (DAPI) (Beijing Zhongshan Jinqiao Biotechnology Co., Ltd., Beijing, China).

### 2.4. Western Blot

Mice were anesthetized with intraperitoneal injection of sodium pentobarbital (35 mg/kg body weight) and euthanized by decapitation. The spinal trigeminal nucleus caudalis (Sp5C) samples were collected on ice. After washing with pre-chilled sterile saline, the Sp5C samples were stored in pre-chilled 1.5 mL low-temperature microtubes and rapidly frozen in liquid nitrogen. The cells were lysed using radio immunoprecipitation assay (RIPA) lysis buffer, followed by ultrasonic disruption. Total protein was collected after centrifugation at 12,000 rpm for 20 min at 4 °C, and the sample volume was calculated. After denaturation, sodium dodecyl sulfate polyacrylamide gel electrophoresis (SDS-PAGE) was performed, followed by transfer to a membrane. The membrane was blocked with 5% non-fat milk and washed with Tris-Buffered Saline with Tween 20 (TBST). Subsequently, the membrane was incubated overnight at 4 °C with primary antibody anti-α-2A adrenergic receptor (α-2AAR) (1:500, YT0298, Imminoway, Suzhou, China). After washing with TBST, the membrane was incubated with goat anti-rabbit immunoglobulin G-horseradish peroxidase (IgG-HRP) (1:10,000, ab6721, Abcam) at room temperature for 2 h. Chemiluminescent detection was performed using an enhanced chemiluminescence (ECL) kit, and β-actin was used as the internal control. Exposure was carried out using an imaging system, and the band intensity was quantified using Image-Pro Plus 6.0 software. The relative expression level of the target protein was calculated as the ratio of the target protein to the internal control protein intensity.

### 2.5. Fiber Photometry Recording

Mice were anesthetized with 0.3% sodium pentobarbital and then fixed in a stereotaxic apparatus. The skin was incised, and the cranial fascia was removed to expose the bregma and lambda. According to the coordinates for the spinal trigeminal nucleus caudalis (Sp5C) region in the mouse brain atlas (AP: −7.64 mm, ML: −1.5 mm, DV: −4.75 mm), a 0.8 mm hole was drilled. A microinjection syringe equipped with a glass electrode filled with paraffin was used to extract 500 nl of the virus (rAAV-hSyn-GRAB-190 NE2m(3.1)-WPRE-pA, BrainVTA (Wuhan) Co., Ltd., Wuhan, China) from the virus reagent tube. Then the virus was injected into Sp5C at a rate of 40 nL/min. After the injection, the optic fiber (diameter of 1.25 mm, 0.1 mm above the site of virus injection, Inper Ltd., Hangzhou, China) was implanted. Once the bone surface dried, 1454 glue and dental cement were applied to fix the fiber optic, and the holder was removed after it hardened. Postoperatively, the mice were allowed to recover on a heating pad. After 3 weeks, fiber-optic recording experiments were conducted under isoflurane (Reward Life Science Technology Co., Ltd., Shenzhen, China) anesthesia. A fiber-optic recording system (ThinkerTech Nanjing BioScience Inc., Nanjing, China) was used to measure the norepinephrine (NE) signal in Sp5C, with the light intensity set at 30–40 μW. Then, the fiber optic was connected to the mouse, and taVNS intervention was applied. The values of NE release change (ΔF/F) from −2 to 5 s (0 s represent the onset of transcutaneous auricular vagus nerve stimulation (taVNS) and were calculated as (F − F0)/F0 for each trial, where F is the NE signal in each image and F0 is defined as the mean of the NE signal before taVNS, subtracted by autofluorescence background. Typical traces were formed using a custom MATLAB(R2024a) script that was developed by ThinkerTech (Hangzhou, China).

### 2.6. Statistical Analysis

Statistical analyses were performed using GraphPad Prism 8 software. Measurement data conforming to a normal distribution were expressed as mean ± standard deviation (Mean ± SD). Group comparisons for behavioral tests, immunofluorescence, and Western blot were conducted using one-way analysis of variance (ANOVA). Group comparisons for fiber-optic recordings were performed using the *t*-test.

Post hoc comparisons were performed using the Least Significant Difference (LSD) test for data exhibiting homogeneous variances. For data with heterogeneous variances, the Games–Howell test was employed. Non-normally distributed data were analyzed using nonparametric methods, with post hoc analyses conducted via the Kruskal–Wallis test. A significance level of *p* ≤ 0.05 was considered statistically significant.

## 3. Results

### 3.1. Comparison of Behavioral Experimental Results Among Groups of Mice

The results of the orofacial stimulation test (OST) in mice showed that there was no difference in drinking time between the groups before the electrical stimulation intervention. After the intervention, compared to the control group, the drinking time of the model group decreased significantly (*p* < 0.001). Compared to the model group, the drinking time of the transcutaneous auricular vagus nerve stimulation (taVNS) group increased significantly (*p* < 0.001). Compared to the taVNS group, the drinking time of the sham taVNS group decreased significantly (*p* < 0.05, Figure 2A).

The results of the mouse grimace scale (MGS) showed that after the electrical stimulation intervention, the facial expression score of the model group increased significantly compared to the control group (*p* < 0.001). Compared to the model group, the facial expression score of the taVNS group decreased significantly (*p* < 0.01). Compared to the taVNS group, the facial expression score of the sham taVNS group showed an increasing trend, but there was no significant difference (*p* > 0.05, Figure 2B).

### 3.2. Comparison of the Number of Norepinephrine (NE) Neurons in the Locus Coeruleus (LC) Among the Groups of Mice

The results of immunofluorescence showed that compared to the control group, the number of NE neurons in the model group decreased significantly (*p* < 0.001). Compared to the model group, the number of NE neurons in the taVNS group increased significantly (*p* < 0.001). Compared to the taVNS group, the number of NE neurons in the sham taVNS group decreased significantly (*p* < 0.001, Figure 3B).

### 3.3. Comparison of NE Release in the Spinal Trigeminal Nucleus Caudalis (Sp5C) Among the Groups of Mice

The fiber-optic recording results showed that the dashed line represents the start of transcutaneous auricular vagus nerve stimulation (taVNS) intervention. The control group exhibited slight fluctuations in fluorescence signal within 5 s of taVNS intervention, while the model group showed a significant increase in fluorescence signal within 5 s. The area under the curve for the two groups showed a significant difference (*p* < 0.001, Figure 4G).

### 3.4. Comparison of α-2A Adrenergic Receptor (α-2AAR) Protein Expression in the Sp5C Among the Groups of Mice

The results of Western blot showed that compared to the control group, the expression of α-2AAR protein in the model group decreased significantly (*p* < 0.01). Compared to the model group, the expression of α-2AAR protein in the taVNS group increased significantly (*p* < 0.001). Compared to the taVNS group, the expression of α-2AAR protein in the sham taVNS group decreased significantly (*p* < 0.01, Figure 5).

## 4. Discussion

The purpose of this study is to reveal the analgesic effect of taVNS on migraine model mice and the role of the locus coeruleus–norepinephrine (LC-NE) system in the vagus central nervous system. Existing clinical results showed that repetitive transcutaneous auricular vagus nerve stimulation (taVNS) could significantly alleviate the symptoms of migraine patients [11]. Our research also showed that after multiple taVNS interventions, the drinking time of the orofacial stimulation test (OST) in migraine mice was significantly prolonged, and the mouse grimace scale (MGS) was significantly decreased, indicating that multiple taVNS could significantly alleviate the pain symptoms of migraine mice.

Patients with migraine showed an abnormally increased functional connectivity of the locus coeruleus (LC) by resting-state functional magnetic resonance imaging and functional connectivity methods [24]. There is evidence of the efficacy of serotonin–norepinephrine reuptake inhibitors, which may be the most effective treatments for patients with depression and migraine [25]. The LC is a dense nucleus composed of closely arranged neurons located above the junction of the pons and the medulla of the brainstem and is regarded as the main source of NE synthesis in the brain. Both single and multiple taVNS can significantly regulate the functional connectivity between the LC and other nuclei in patients with migraine [16,17]. Both 1 Hz and 20 Hz taVNS can regulate the functional connectivity between the LC and other brain regions in different ways [18]. Therefore, based on the parameters of the Hans-200A electroacupuncture device in our laboratory, 2/15 Hz taVNS was applied in the experiment. Different intensities of taVNS have different activation effects on the LC. While volunteers received low-intensity taVNS to the left external acoustic meatus, the pupil size was recorded during a pupillary light reflex task, but not for participants who received higher-intensity taVNS [26]. So, we adopted 1 mA taVNS for migraine mice.

The auricular branch of the vagus nerve supplies input to multiple nuclei in the brainstem, which then activate the LC to regulate the level of norepinephrine (NE). Numerous studies have also reported that taVNS targets and modulates the NE system in the LC [27]. The LC-NE system may play an important role in the taVNS mechanism. Therefore, we investigated the modulatory effects of taVNS on the LC-NE system in migraine mice. Our results showed that taVNS significantly increased the expression of NE in the LC, indicating that taVNS activated the NE system in migraine mice.

The neural pathways and structures that are controlled and activated by the NE system are complex. The LC-NE system regulates various brain functions by releasing NE projections to different regional targets. Repeated taVNS significantly regulated the resting-state functional connectivity between the LC and brain regions that are closely related to pain processing and modulation, which was significantly associated with a reduction in the number of migraine days [17]. The neurons of the LC have a complex ascending and descending projection network. The descending axons of NE neurons target the spinal trigeminal nucleus caudalis (Sp5C) and the pain pathway of the entry region of the trigeminal nerve, which provides endogenous inhibitory control of nociceptive pain [28]. Pickering and colleagues reported that silencing NE neurons using an adeno-associated virus vector induced heat hyperalgesia and increased inflammation-associated heat hypersensitivity caused by plantar injection of complete Freund’s adjuvant injection [29]. Our results showed that taVNS significantly increased the release of NE in the Sp5C and upregulated the α-2A adrenergic receptor(α-2AAR) (Figure 6). A large number of basic studies have shown that the inhibition of NE on noxious transmission in the spinal cord is one of the most important pain suppression systems emitted from the brainstem [30]. It is indicated that taVNS may activate the NE system and exert a descending inhibitory effect on Sp5C through α-2AAR, thereby alleviating the occurrence of migraines (Figure 6).

Serotonin norepinephrine reuptake inhibitors (SNRIs), including venlafaxine and duloxetine, have been shown to have efficacy and may be the most effective treatments in patients with comorbid depression and migraine [25]. Our research showed that taVNS exerts analgesic effects on migraine through the downward inhibitory pathway of the NE system. This indicates that taVNS also shows great potential in pain and depression comorbidities.

In addition, our results also showed that both taVNS and sham taVNS (ear edge stimulation) stimulated the relieving of symptoms of migraine to varying degrees, but they had different regulatory effects on the LC^NE^–Sp5C^α-2AAR^. Another study also demonstrated that both taVNS and earlobe stimulation improved alertness levels following sleep deprivation, but the possible mechanisms involved may be different. taVNS significantly increased the salivary alpha-amylase level, indicating activation of the LC-NE system, whereas earlobe stimulation significantly increased the heart rate and reduced heart rate variability, promoting sympathetic nerve activity [31]. Therefore, the central mechanisms of the effect of sham taVNS on migraine mice remains to be further studied.

This study provides insights into the neural mechanism of taVNS. Future research can elucidate the mechanism of action of taVNS by examining the interactions between higher-level central nuclei, and preliminary clinical studies could be carried out to validate its efficacy and safety in humans, including optimizing the stimulation parameters. This study also provided a promising intervention for migraine in clinical practice.

In conclusion, we observed that repeated taVNS can modulate key nodes in the brainstem–vagus nerve pathway, where the LC and its NE system exert a descending inhibitory analgesic effect on the Sp5C. This may represent the central mechanism through which taVNS alleviates migraines.

## Figures and Tables

**Figure 1 biomedicines-14-00096-f001:**
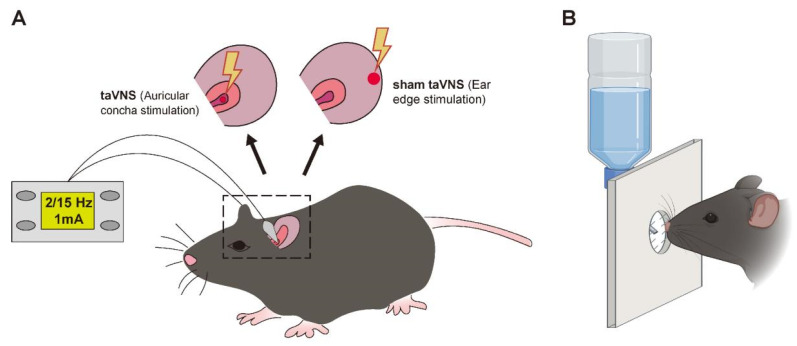
Schematic of taVNS, sham taVNS, and OST. (**A**) Schematic of taVNS and sham taVNS. (**B**) Schematic of OST.(Created in Biorender. Xingke Song (2025) https://app.biorender.com/illustrations/681f8d35cb9cc2dd8818fd1b).

**Figure 2 biomedicines-14-00096-f002:**
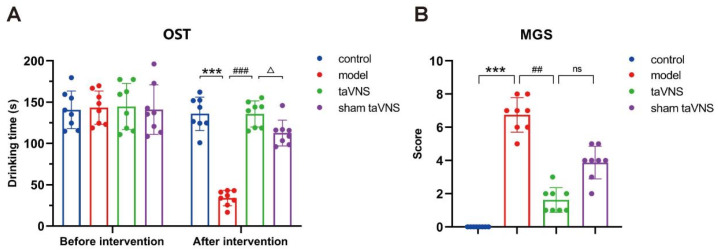
Comparison of behavioral experiment results among groups of mice. Data are presented as mean ± standard deviation (Mean ± SD), with *n* = 8 per group. (**A**) OST, *** *p* < 0.001; ### *p* < 0.001; △ *p* < 0.05. (**B**) MGS, *** *p* < 0.001; ## *p* < 0.01.

**Figure 3 biomedicines-14-00096-f003:**
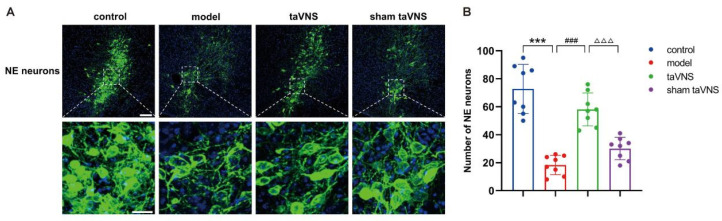
Comparison of the number of NE neurons in the LC among the groups of mice. Data are presented as mean ± standard deviation (Mean ± SD), with *n* = 8 per group. (**A**) Immunofluorescence of NE neurons in different groups of mice. Scale bars: upper 100 μm, lower 10 μm. (**B**) The number of NE neurons in the LC n different groups of mice, *** *p* < 0.001; ### *p* < 0.001; △△△ *p* < 0.001.

**Figure 4 biomedicines-14-00096-f004:**
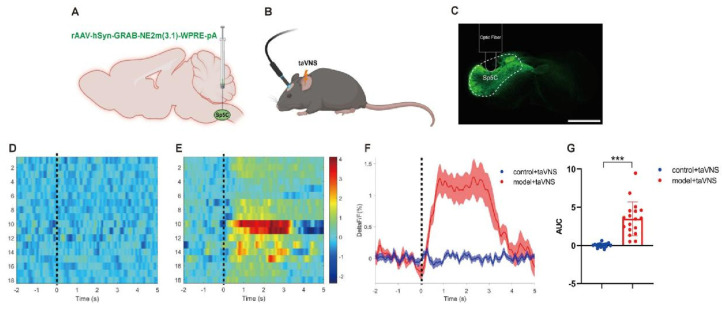
Comparison of NE release in the Sp5C among the groups of mice. Data are presented as mean ± standard deviation (Mean ± SD), with *n* = 6 per group. (**A**) AAV−NE2m injected into Sp5C. (**B**) Schematic of fiber photometry recording. (Created in Biorender. Xingke Song (2025) https://app.biorender.com/illustrations/6820da1835a71c59a5fe9b24) (**C**) Representative image showing expression of NE2m in Sp5C. Scale bar, 1000 μm.(**D**) Heatmap of average ΔF/F signal when the control + taVNS group received taVNS intervention (values of average ΔF/F signal were averages of 3 trials for each animal). (**E**) Heatmap of average ΔF/F signal when the model + taVNS group received taVNS intervention (values of average ΔF/F signal were averages of 3 trials for each animal). (**F**) Example trace of NE release in Sp5C in the model + taVNS group (upper) and control + taVNS group (lower). (**G**) Area under the curve of average ΔF/F signal between the two groups, *** *p* < 0.001.

**Figure 5 biomedicines-14-00096-f005:**
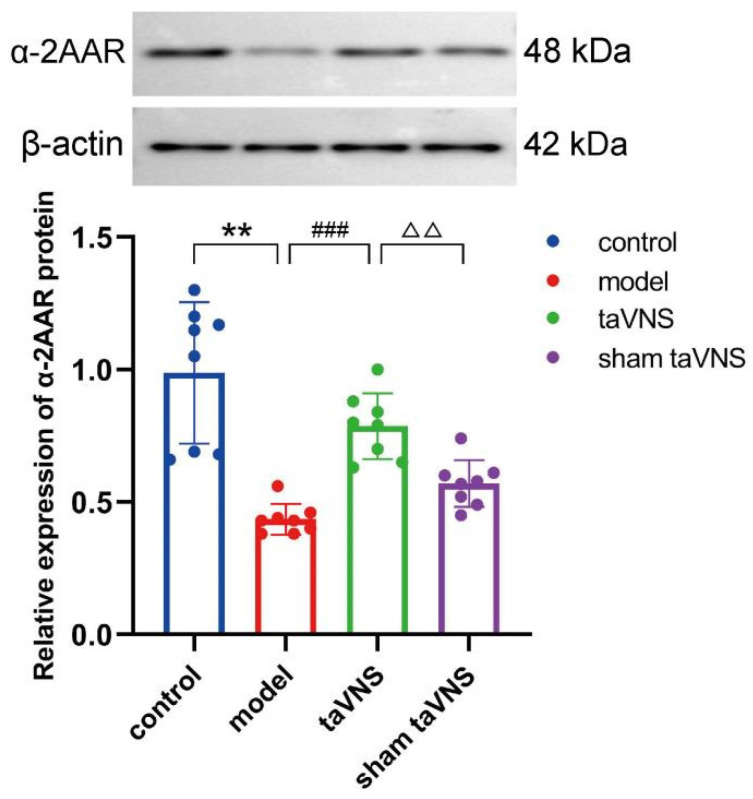
Comparison of relative expression of α-2AAR protein in the Sp5C among the groups of mice. Data are presented as mean ± standard deviation (Mean ± SD), with *n* = 8 per group, ** *p* < 0.01; ### *p* < 0.001; △△ *p* < 0.01.

**Figure 6 biomedicines-14-00096-f006:**
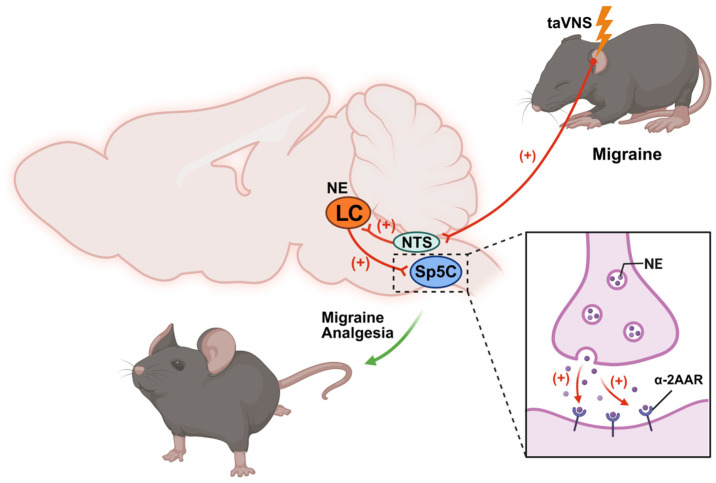
Diagram of the analgesic mechanism of taVNS intervention in migraine mice. taVNS is capable of inducing electrical activity in the vagus nerve, with afferent impulses being transmitted via the NTS to the LC. This activation facilitates the release of NE from LC neurons, which subsequently binds to α-2AAR on neurons within the Sp5C, thereby mediating inhibition of nociceptive transmission and exerting analgesic effects on migraine. (Created in Biorender. Xingke Song (2025) https://app.biorender.com/illustrations/6854b7ec43b2f62cc9d7a723).

**Table 1 biomedicines-14-00096-t001:** The mouse grimace scale.

Score	0	1	2
Eyes (degree of eye opening)	Eyes fully open	Eyes partially closed	Eyes fully closed or squinted
Nose (nasal ridge)	Smooth nasal ridge	Slightly raised nasal ridge	Prominent nasal ridge
Cheeks (cheek puffing)	Flat cheeks	Slightly puffed cheeks	Clearly puffed cheeks
Ears (orientation to the sides)	Ears naturally relaxed	Ears tilted outward	Increased gap between the ears
Whiskers (upright)	Whiskers relaxed, slightly drooping	Whiskers tilted backward	Whisker pads contracted, whiskers upright

## Data Availability

The datasets presented in this article are not readily available, but the data that support the findings of this study are available on reasonable request to the corresponding author. Requests to access the datasets should be directed to Junying Wang, wjyanguning@aliyun.com.

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
