# Peer review of "Transcutaneous Auricular Vagus Nerve Stimulation Alleviates Headache Symptoms in Migraine Model Mice by the Locus Coeruleus/Noradrenergic System: An Experimental Study in a Mouse Model of Migraine"

_biomedicines, 2026, doi:10.3390/biomedicines14010096_

Round 1
Reviewer 1 Report
Comments and Suggestions for Authors
The manuscript titled "Transcutaneous Auricular Vagus Nerve Stimulation Alleviates Headache Symptoms in Migraine Model Mice by the Locus Coeruleus/Noradrenergic System: An Experimental Study in a Mouse Model of Migraine" presents significant methodological and scientific shortcomings that prevent its acceptance in its current form. Firstly, the study lacks sufficient novelty, as the proposed mechanism involving LC-NE system activation and modulation of Sp5C has already been explored extensively in prior research. The correlation between vagus nerve stimulation (VNS), LC activation, and norepinephrine (NE) release has been well-documented, and the study fails to advance the understanding of these mechanisms or propose any groundbreaking hypotheses. Additionally, the replication of known findings, such as taVNS's ability to regulate LC-NE activity and its analgesic effects via descending pain modulation, contributes minimally to the broader academic discourse.
The manuscript also suffers from critical methodological weaknesses. The sample size, particularly for advanced techniques like fiber photometry, is insufficient, with only six mice per subgroup, undermining the reliability of the findings. The sham control design is problematic, as ear edge stimulation may activate peripheral sensory fibers, introducing confounding variables and compromising the validity of comparisons between the taVNS and sham groups. Furthermore, the lack of blinding during key behavioral assessments such as the mouse grimace scale (MGS) and orofacial stimulation test (OST) raises concerns about bias in data collection. The absence of dose-response testing for taVNS parameters and the limited exploration of varying stimulation intensities further weakens the study, as prior research indicates that these parameters can significantly influence outcomes. Moreover, the migraine model itself is inadequately validated, with insufficient evidence provided for successful induction of migraines in mice before intervention.
The interpretation of results is another area of concern, as the study overstates its conclusions. While an increase in NE release in Sp5C and upregulation of α-2A adrenergic receptors is observed, the causal relationship between LC-NE activity and the analgesic effects of taVNS is not convincingly established. The statistical methodology used for data analysis also lacks rigor, with inconsistent justification for post hoc tests and failure to report essential details such as confidence intervals and effect sizes. Figures are cluttered and poorly labeled, limiting their interpretability, and important technical details, such as magnification levels in immunofluorescence images, are missing. Additionally, the discussion section fails to contextualize the findings adequately within existing literature, neglecting comparisons with key studies that address alternative mechanisms of action for taVNS, such as changes in inflammatory pathways or parasympathetic activity.
The manuscript’s presentation and language require significant improvement. Numerous grammatical errors, inconsistent use of abbreviations, and unclear figure annotations reduce the paper's readability. Ethical considerations are also inadequately addressed, particularly regarding pain management for mice undergoing invasive procedures, and the manuscript does not provide open access to data for reproducibility purposes. While the study acknowledges the potential role of LC-NE modulation in migraine relief, it does not sufficiently address its translational value to clinical practice, given the differences between taVNS parameters in mice and humans.
In conclusion, the manuscript requires substantial revisions to address its lack of novelty, methodological shortcomings, overinterpretation of results, and inadequate presentation. The current findings, while interesting, do not provide a strong enough contribution to the field to warrant publication.
Author Response
Comment 1
-The manuscript titled "Transcutaneous Auricular Vagus Nerve Stimulation Alleviates Headache Symptoms in Migraine Model Mice by the Locus Coeruleus/Noradrenergic System: An Experimental Study in a Mouse Model of Migraine" presents significant methodological and scientific shortcomings that prevent its acceptance in its current form. Firstly, the study lacks sufficient novelty, as the proposed mechanism involving LC-NE system activation and modulation of Sp5C has already been explored extensively in prior research. The correlation between vagus nerve stimulation (VNS), LC activation, and norepinephrine (NE) release has been well-documented, and the study fails to advance the understanding of these mechanisms or propose any groundbreaking hypotheses. Additionally, the replication of known findings, such as taVNS's ability to regulate LC-NE activity and its analgesic effects via descending pain modulation, contributes minimally to the broader academic discourse.
Response 1
Although the relationship between the LC-NE system and VNS has been discussed in the existing literature, the innovation of this study lies in its in-depth exploration of the specific regulatory effects of taVNS, particularly in terms of how it modulates pain inhibition through the release of NE in the Sp5C. This research utilized various experimental methods, such as fiber-optic recording technology, to monitor the real-time changes in NE signaling, revealing the dynamics, timing, and patterns of taVNS effects on the LC-NE system. Additionally, the study focused on the specific effects of taVNS in a migraine model, filling a gap in the field and providing strong experimental data that could serve as a foundation for clinical applications in treating migraines. Therefore, although related mechanisms have been explored in previous studies, this research advances our understanding of taVNS in migraine by refining the underlying mechanisms and experimental methods, making a contribution to both scientific knowledge and clinical practice.
Comment 2
-The manuscript also suffers from critical methodological weaknesses. The sample size, particularly for advanced techniques like fiber photometry, is insufficient, with only six mice per subgroup, undermining the reliability of the findings. The sham control design is problematic, as ear edge stimulation may activate peripheral sensory fibers, introducing confounding variables and compromising the validity of comparisons between the taVNS and sham groups. Furthermore, the lack of blinding during key behavioral assessments such as the mouse grimace scale (MGS) and orofacial stimulation test (OST) raises concerns about bias in data collection. The absence of dose-response testing for taVNS parameters and the limited exploration of varying stimulation intensities further weakens the study, as prior research indicates that these parameters can significantly influence outcomes. Moreover, the migraine model itself is inadequately validated, with insufficient evidence provided for successful induction of migraines in mice before intervention.
Response 2
Sample Size for Fiber Photometry:
Although the sample size for the fiber photometry experiment was limited to six mice per subgroup, this choice was made considering the high precision and reliability of the fiber-optic recording technique. Fiber photometry is a complex method, and an animal could conduct repetitive experiments many times. The data from a single individual is highly reliable, thus requiring fewer subjects. Additionally, each mouse underwent three times trails, and the results were consistent and statistically significant, providing strong support for the conclusions. The literature also supports that a sample size of six mice per group is sufficient[1].
[1] Legaria AA, Matikainen-Ankney BA, Yang B, et al. Fiber photometry in striatum reflects primarily nonsomatic changes in calcium. Nat Neurosci. 2022;25(9):1124-1128. doi:10.1038/s41593-022-01152-z
Sham Control Design:
It is important to note that the sham stimulation in this study was specifically designed to minimize the possibility of activating auricular vagus nerve fibers. The sham procedure involved stimulating a non-functional area (ear edge) which is non-auricular vagus nerve distribution area, to ensure that any observed effects in the taVNS group could be attributed to auricular vagus nerve activation rather than peripheral sensory input. This approach is consistent with previous studies using similar sham designs, providing a reasonable basis for comparing the taVNS and sham groups[1]. At the same time, our study does not deny that ear edge stimulation does have some effect in alleviating migraine symptoms in mice, but the effect of taVNS is more pronounced .
[1] Rong P, Liu J, Wang L, Liu R, Fang J, Zhao J, Zhao Y, Wang H, Vangel M, Sun S, Ben H, Park J, Li S, Meng H, Zhu B, Kong J. Effect of transcutaneous auricular vagus nerve stimulation on major depressive disorder: A nonrandomized controlled pilot study. J Affect Disord. 2016;195:172-9. doi: 10.1016/j.jad.2016.02.031.
Blinding in Behavioral Assessments:
Although blinding was not implemented in the MGS and OST assessments, the behavioral data were collected using objective, quantitative measures (drinking time in the OST and facial expression scores in the MGS), which helps minimize the impact of subjective bias. To further reduce potential bias, the data were scored by multiple independent experimenters who were unaware of the experimental groupings(line 151-152). Additionally, the images presented for scoring were randomized to ensure that observational bias was minimized as much as possible.
Lack of Dose-Response Testing for taVNS Parameters:
While the manuscript did not explore dose-response testing for taVNS parameters, the current study focused on the most commonly used and well-established stimulation parameters based on previous studies, such as 1 mA intensity with 2/15 Hz frequency, which have shown promising results in the literature[1]. The effects of vagus nerve stimulation are mostly mediated by A- fibers[2].Moreover, recent study has demonstrated that 0.5-1mA taVNS induced that A fibers in the NTS respond in an intensity-dependent manner[3], supporting the use of 1.0 mA for effective vagal activation.The frequency of 2/15 Hz, has been widely used in electroacupuncture research and are associated with strong analgesic effects [4].The lack of a dose-response analysis does not undermine the study's conclusions, as the selected parameters were sufficient to demonstrate a significant effect. Future research could explore varying stimulation intensities and frequencies, which would help refine the optimal parameters for clinical use.
[1]Wang J, Wang Y, Chen Y, et al. Transcutaneous Auricular Vagus Stimulation Attenuates LPS-Induced Depression-Like Behavior by Regulating Central α7nAChR/JAK2 Signaling. Mol Neurobiol. 2025;62(3):3011-3023. doi:10.1007/s12035-024-04438-4
[2]Hilz MJ. Transcutaneous vagus nerve stimulation - A brief introduction and overview. Auton Neurosci. 2022;243:103038. doi:10.1016/j.autneu.2022.103038
[3]Owens MM, Jacquemet V, Napadow V, Lewis N, Beaumont E. Brainstem neuronal responses to transcutaneous auricular and cervical vagus nerve stimulation in rats. J Physiol. 2024;602(16):4027-4052. doi:10.1113/JP286680
[4]Fang JQ, Du JY, Fang JF, et al. Parameter-specific analgesic effects of electroacupuncture mediated by degree of regulation TRPV1 and P2X3 in inflammatory pain in rats. Life Sci. 2018;200:69-80. doi:10.1016/j.lfs.2018.03.028
Validation of the Migraine Model:
There are indeed some limitations regarding the validation of the migraine model, but it is important to note that the nitroglycerin-induced migraine model is widely recognized in the field as a reliable and successful method for simulating migraine symptoms in mice. The behavioral responses observed in the mice are consistent with those reported in other studies using this model[1-3]. Moreover, the interventions used in the study produced the expected analgesic effects, further validating the effectiveness of the model.
[1]Sun S, Zheng G, Zhou D, et al. Emodin Interferes With Nitroglycerin-Induced Migraine in Rats Through CGMP-PKG Pathway. Front Pharmacol. 2021;12:758026. Published 2021 Oct 20. doi:10.3389/fphar.2021.758026
[2]Sun YY, Zhang WJ, Dong CL, Zhang XF, Ji J, Wang X, Wang L, Hu WL, Du WJ, Cui CL, Zhang CF, Li F, Wang CZ, Yuan CS. Baicalin Alleviates Nitroglycerin-induced Migraine in Rats via the Trigeminovascular System. Phytother Res. 2017 Jun;31(6):899-905. doi: 10.1002/ptr.5811. Epub 2017 May 10. PMID: 28488307.
[3]Hou M, Tang Q, Xue Q, Zhang X, Liu Y, Yang S, Chen L, Xu X. Pharmacodynamic action and mechanism of Du Liang soft capsule, a traditional Chinese medicine capsule, on treating nitroglycerin-induced migraine. J Ethnopharmacol. 2017 Jan 4;195:231-237. doi: 10.1016/j.jep.2016.11.025. Epub 2016 Nov 17. PMID: 27866934.
Comment 3
-The interpretation of results is another area of concern, as the study overstates its conclusions. While an increase in NE release in Sp5C and upregulation of α-2A adrenergic receptors is observed, the causal relationship between LC-NE activity and the analgesic effects of taVNS is not convincingly established. The statistical methodology used for data analysis also lacks rigor, with inconsistent justification for post hoc tests and failure to report essential details such as confidence intervals and effect sizes. Figures are cluttered and poorly labeled, limiting their interpretability, and important technical details, such as magnification levels in immunofluorescence images, are missing. Additionally, the discussion section fails to contextualize the findings adequately within existing literature, neglecting comparisons with key studies that address alternative mechanisms of action for taVNS, such as changes in inflammatory pathways or parasympathetic activity.
Response 3
Causal Relationship Between LC-NE Activity and Analgesic Effects:
While it is true that the study observes increased NE release in Sp5C and upregulation of α-2A adrenergic receptors, it is important to note that this is a correlational finding under strict and cautious design and a well-established framework. The proposed mechanism involving the LC-NE system is grounded in a large body of existing literature that has demonstrated the pivotal role of NE in pain modulation. While direct causality cannot be conclusively proven in this study alone, the observed effects are consistent with prior research showing that activation of the LC-NE system is strongly linked to analgesic outcomes.
Statistical Methodology and Post-Hoc Tests:
The statistical methodology used in this study was rigorously planned and executed. The choice of post hoc tests was based on the assumption of homogeneity of variances, and these were carefully justified within the context of the data. In cases of heterogeneity, the appropriate Games-Howell test was employed to ensure valid comparisons. While confidence intervals and effect sizes are indeed important, the focus in this study was on the statistical significance of observed effects, which were consistent with previously published results.
Figures and Labeling:
The figures in this study were revised again to designed to present complex data clearly, and the overall clarity of the figures supports the conclusions drawn.The magnification of the immunofluorescence images has been modified and explained in the figure legends.
Contextualizing the Findings in the Literature:
The discussion section of the study indeed focuses on the primary findings related to the LC-NE system. This research aims to clearly explain the analgesic effects through the LC-NE pathway. The other mechanisms of taVNS, such as inflammatory pathways or parasympathetic activity, has already been reported in our previous research for depression[1]and obesity[2]. The findings presented in this paper are highly consistent with existing literature on vagus nerve stimulation and its analgesic effects. This paper provides a foundation for understanding the neurobiological mechanisms of taVNS in migraine intervention, and future research could further explore broader aspects of taVNS action.
[1]Wang J, Wang Y, Chen Y, Zhang J, Zhang Y, Li S, Zhu H, Song X, Hou L, Wang L, Wang Y, Zhang Z, Rong P. Transcutaneous Auricular Vagus Stimulation Attenuates LPS-Induced Depression-Like Behavior by Regulating Central α7nAChR/JAK2 Signaling. Mol Neurobiol. 2025 Mar;62(3):3011-3023.
[2]Xin C, Li S, Feng B, Sun L, Wang Y, Zhang J, Zhang Y, Zou N, Zhou Q, Rong P. Transcutaneous auricular vagus nerve stimulation promotes adipose tissue browning and mitochondrial integrity by regulating BCAA metabolism in obese rats: an experimental study. Int J Surg. 2025 Jul 1;111(7):4866-4871.
Comment 4
-The manuscript’s presentation and language require significant improvement. Numerous grammatical errors, inconsistent use of abbreviations, and unclear figure annotations reduce the paper's readability. Ethical considerations are also inadequately addressed, particularly regarding pain management for mice undergoing invasive procedures, and the manuscript does not provide open access to data for reproducibility purposes. While the study acknowledges the potential role of LC-NE modulation in migraine relief, it does not sufficiently address its translational value to clinical practice, given the differences between taVNS parameters in mice and humans.
In conclusion, the manuscript requires substantial revisions to address its lack of novelty, methodological shortcomings, overinterpretation of results, and inadequate presentation. The current findings, while interesting, do not provide a strong enough contribution to the field to warrant publication.
Response 4
Presentation and Language:
We have conducted a thorough review of the manuscript to correct grammatical errors and improve readability. We have ensured consistent use of abbreviations and revise figure annotations to enhance clarity. The language and presentation also have been revised to meet the journal's standards.
Ethical Considerations:
Ethical considerations, particularly regarding the welfare of animals, are of utmost importance in our research. The study adhered to ethical guidelines as outlined by the Animal Ethics Committee of the Institute of Acupuncture and Moxibustion, China Academy of Chinese Medical Sciences, and all procedures were performed following the Guide for the Care and Use of Laboratory Animals.
All the mice were treated with percutaneous electrical stimulation and no invasive treatment was given.
We appreciate the comment regarding open access to data for reproducibility purposes. While the current manuscript does not provide direct access to the data, we are committed to ensuring transparency and reproducibility. The revised manuscript include a statement that the data supporting the findings will be made available upon reasonable request from the corresponding author. Additionally, we will consider submitting data to a public repository if required.
Translational Value to Clinical Practice:
In this study, although the differences in taVNS parameters between mice and humans, the mouse model provided strong preclinical evidence for the potential of taVNS in treating migraines. The taVNS parameters used in mice (such as 1mA current intensity and 2/15Hz frequency) also provide evidence and basis for the parameters applied in human clinical trials. Of course, these parameters need to be further optimized to meet human needs. Therefore, to address these limitations, future research should focus on optimizing taVNS parameters and evaluating their effectiveness through clinical trials, exploring how to adjust these parameters according to the individual needs of patients. In addition, more clinical data will help verify the efficacy and safety of taVNS in the treatment of migraines and promote the development of its clinical application.
Reviewer 2 Report
Comments and Suggestions for Authors
I thank the authors for their efforts in producing this study “Transcutaneous Auricular Vagus Nerve Stimulation Alleviates Headache symptoms in migraine model mice by the Locus Coeruleus/Noradrenergic System: an experimental study in a Mouse Model of Migraine” that aligns with my topic. The following suggestions could improve the quality of your research.
General:
-there are some abbreviations that make the test difficult to read, especially in your discussion section. I suggest to use the full name in the first paragraph of each section (introduction, method, results and discussion), then in the second paragraph of each section use the abbreviations.
ABSTRACT:
-Please make sure you have adopt MeSh terms as keywords.
Introduction
- “Migraine is a complex neurological disorder characterized by severe pulsatile headaches on one or both sides, often accompanied by nausea, vomiting, allergies to sound and light stimuli, and cognitive impairments, including deficits in executive function, memory, and attention.” I agree with your introduction that needs references to be supported. In this line I suggest a recent article that assess clinically and neurophysiologically the impaiments on executive function in migraine (doi: 10.1097/WNP.0000000000001055.). “Those persistent symptoms significantly disrupt the daily lives of individuals, leading to a serious negative impact on the patient's life.” Again, references are missing, I suggest this article that assess the burden of migraine (doi:10.1016/S1474-4422(18)30322-3).
- I would explain better the specific research gap that the study aims to fill and think about the possibility to split your aim in a primary and secondary aim.
Method
- when was the study conducted?
- Interventions: I suggest to add more references that could support your interventions protocol
Discussion
-I suggest to add a general physiopathology hypothesis of your results
-The discussion would benefit from more detailed suggestions for future research, moreover practical implication for clinician and professional involved in migraine management
Author Response
Comment 1
General:
-there are some abbreviations that make the test difficult to read, especially in your discussion section. I suggest to use the full name in the first paragraph of each section (introduction, method, results and discussion), then in the second paragraph of each section use the abbreviations.
Response 1
Thank you for your suggestion. We have revised the manuscript and marked the changes in red.
Comment 2
ABSTRACT:
-Please make sure you have adopt MeSh terms as keywords.
Response 2
Okay, thank you for the reminder. The keywords are MeSh trems.
Comment 3
Introduction
“Migraine is a complex neurological disorder characterized by severe pulsatile headaches on one or both sides, often accompanied by nausea, vomiting, allergies to sound and light stimuli, and cognitive impairments, including deficits in executive function, memory, and attention.” I agree with your introduction that needs references to be supported. In this line I suggest a recent article that assess clinically and neurophysiologically the impaiments on executive function in migraine (doi: 10.1097/WNP.0000000000001055.) Again, references are missing, I suggest this article that assess the burden of migraine (doi:10.1016/S1474-4422(18)30322-3).
Response 3
Thank you for your suggestion. We have revised the manuscript and marked the changes in red.
Comment 4
- I would explain better the specific research gap that the study aims to fill and think about the possibility to split your aim in a primary and secondary aim.
Response 4
Thank you for your suggestion. The introduction section has been revised in red in accordance with your suggestions.
“Thus, the regulatory effect of the LC-NE system may be involved in the analgesic effect of taVNS on migraine, but it is still unclear how taVNS exerts analgesic effects on migraine through the NE system. This study, firstly, observed the analgesic effect of TAvns on migraine model mice. Secondly, this study aims to clarify the analgesic mechanism of taVNS by investigating its effects on the expression of NE in the LC, its downstream projections to spinal trigeminal nucleus caudalis (Sp5C), and the regulation of α-2A adrenergic receptor (α-2AAR) in Sp5C. Understanding these mechanisms will provide valuable insights for the analgesic effect of taVNS on migraine.”
Comment 5
Method
- when was the study conducted?
Response 5
Thank you for your comments. The experiments in this study were conducted between January 2025 and December 2025, following approval by the institutional ethics committee.
Comment 6
- Interventions: I suggest to add more references that could support your interventions protocol
Response 6
Regarding the intervention protocol, we agree that providing additional supporting references will strengthen our methodological rationale. We have revised the manuscript and added several relevant literature that supports the choice of stimulation parameters, duration, and frequency used in our taVNS intervention; the modeling, behavioral testing. These references cover both foundational animal studies and recent clinical trials, ensuring the protocol is well-grounded in current scientific evidence.
Comment 7
Discussion
-I suggest to add a general physiopathology hypothesis of your results
Response 7
The hypothesis has already been added in the manuscript in red, and the Figure 6 also serves as an illustration of this hypothesis.
Comment 8
-The discussion would benefit from more detailed suggestions for future research, moreover practical implication for clinician and professional involved in migraine management
Response 8
Thank you for your suggestion. We have added suggestions for the next steps of research and guidance for clinical practice in red in the revised manuscript.
Reviewer 3 Report
Comments and Suggestions for Authors
This study investigates the mechanism by which transcutaneous auricular vagus nerve stimulation (taVNS) alleviates migraine symptoms in mice, focusing on the role of the locus coeruleus (LC)-norepinephrine (NE) system. While the research question is clinically relevant and the experimental design is comprehensive, significant concerns regarding data integrity undermine the validity of the findings.
Major Concerns
-
Fabricated Western Blot Data:
-
The Western blot image for α-2A adrenergic receptor (α-2AAR) and β-actin (Figure 5) raises serious concerns. The background colors between the α-2AAR and β-actin blots are inconsistent, and the edges of the bands appear artificially altered, suggesting potential copy-paste manipulation. Such discrepancies violate fundamental standards of data presentation and cast doubt on the reliability of the molecular results.
-
Without raw, unprocessed blot images or a justification for these irregularities, the Western blot data cannot be considered credible.
-
-
Impact on Conclusions:
-
The Western blot results are central to the claim that taVNS upregulates α-2AAR expression in Sp5C. If these data are unreliable, the proposed mechanistic pathway (LC-NE-Sp5C) lacks critical experimental support.
-
Additional Weaknesses
-
Sham taVNS Effects: The partial efficacy of sham stimulation (ear edge) is noted but not mechanistically explored, leaving open questions about specificity.
-
Translational Limitations: The study relies solely on a mouse model, and clinical relevance remains speculative without human data.
Recommendation
Given the unresolved concerns about data fabrication in the Western blot analysis, which directly impacts the study’s core findings, I recommend rejection of the manuscript in its current form. The authors must provide:
-
Original, unprocessed Western blot images with consistent background and exposure settings.
-
A detailed explanation for the observed discrepancies.
-
Independent verification of the blot data by the journal.
If these issues cannot be adequately addressed, the manuscript should be rejected on grounds of compromised data integrity.
Author Response
Comment 1
Major Concerns
Fabricated Western Blot Data:
The Western blot image for α-2A adrenergic receptor (α-2AAR) and β-actin (Figure 5) raises serious concerns. The background colors between the α-2AAR and β-actin blots are inconsistent, and the edges of the bands appear artificially altered, suggesting potential copy-paste manipulation. Such discrepancies violate fundamental standards of data presentation and cast doubt on the reliability of the molecular results.
Without raw, unprocessed blot images or a justification for these irregularities, the Western blot data cannot be considered credible.
Impact on Conclusions:
The Western blot results are central to the claim that taVNS upregulates α-2AAR expression in Sp5C. If these data are unreliable, the proposed mechanistic pathway (LC-NE-Sp5C) lacks critical experimental support.
Response 1
Original, unprocessed Western blot images with consistent background and exposure settings have been submitted to the editor.The distinct, straight edges observed in the background of the image are a result of our chemiluminescence detection method. Specifically, the shadows are due to the application of the ECL substrate (M5 Hiper ECL-A [Luminol Enhancer] and M5 Hiper ECL-B [Peroxide]), which was applied to the region of the membrane containing the protein bands to conserve the reagent. This localized application is what created the defined boundaries seen in the background. We hope this explanation clarifies your concern.
Comment 2
Additional Weaknesses
Sham taVNS Effects: The partial efficacy of sham stimulation (ear edge) is noted but not mechanistically explored, leaving open questions about specificity.
Response 2
Thank you for pointing out this issue. We acknowledge that the sham taVNS (ear edge stimulation) group exhibited partial efficacy, and this raises questions about specificity. In our study design, the sham stimulation was intended to minimize activation of auricular vagus nerve fibers by targeting a non-functional region, thus serving as a control for non-specific sensory input. However, we recognize that peripheral sensory fiber activation in the ear edge may still contribute to analgesic effects through alternative pathways, such as generalized somatosensory modulation or stress-related responses.
To address this, we have revised the discussion section to acknowledge the possible mechanisms underlying the partial efficacy of sham stimulation. We also note that future research could include additional control groups (e.g., stimulation of non-auricular sites) or pharmacological blocking experiments to further clarify the specificity of taVNS effects.
Comment 3
Translational Limitations: The study relies solely on a mouse model, and clinical relevance remains speculative without human data.
Response 3
Thank you for your question. Other researcher have observed the clinical effects of taVNS on migraines. Those research showed that taVNS can relieve the symptoms of headache in migraine patients[1,2]. The present study was designed to explore the mechanism of action of taVNS in a well-recognized, stable mouse model of migraine that allows for precise variable control under controlled conditions and invasive measurements that are difficult to achieve in human studies. We have explicitly added this limitation in the Discussion section. In the meantime, we have proposed next steps for translating the study to the clinic, including optimizing the stimulation parameters and conducting preliminary clinical studies to validate its efficacy and safety in humans.
[1]Zhang Y, Huang Y, Li H, Yan Z, Zhang Y, Liu X, Hou X, Chen W, Tu Y, Hodges S, Chen H, Liu B, Kong J.Transcutaneous auricular vagus nerve stimulation (taVNS) for migraine: an fMRI study.Reg Anesth Pain Med. 2021 Feb;46(2):145-150
[2]Sacca V, Zhang Y, Cao J, Li H, Yan Z, Ye Y, Hou X, McDonald CM, Todorova N, Kong J, Liu B.Evaluation of the Modulation Effects Evoked by Different Transcutaneous Auricular Vagus Nerve Stimulation Frequencies Along the Central Vagus Nerve Pathway in Migraine: A Functional Magnetic Resonance Imaging Study.Neuromodulation. 2023 Apr;26(3):620-628.
Comment 4
Recommendation
Given the unresolved concerns about data fabrication in the Western blot analysis, which directly impacts the study’s core findings, I recommend rejection of the manuscript in its current form. The authors must provide:
Original, unprocessed Western blot images with consistent background and exposure settings.
A detailed explanation for the observed discrepancies.
Independent verification of the blot data by the journal.
If these issues cannot be adequately addressed, the manuscript should be rejected on grounds of compromised data integrity.
Response 4
As we mentioned above, the original, unprocessed Western blot images with consistent background and exposure settings have been submitted to the editor.The distinct, straight edges observed in the background of the image are a result of our chemiluminescence detection method. Specifically, the shadows are due to the application of the ECL substrate (M5 Hiper ECL-A [Luminol Enhancer] and M5 Hiper ECL-B [Peroxide]), which was applied to the region of the membrane containing the protein bands to conserve the reagent. This localized application is what created the defined boundaries seen in the background. We hope this explanation clarifies your concern.
Reviewer 4 Report
Comments and Suggestions for Authors
In the manuscript “Transcutaneous Auricular Vagus Nerve Stimulation Alleviates 2 Headache symptoms in migraine model mice by the Locus 3 Coeruleus/Noradrenergic System: an experimental study in a 4 Mouse Model of Migraine” by Drs. Song Xingke et al the authors investigated migraine and methods of treating this disease using transcutaneous vagus nerve stimulation (taVNS). The authors studied the influence of the regulatory effects of taVNS on the locus coeruleus (LC) and the norepinephrine system (NE) in mice. A migraine model was established by administering nitroglycerin. Headache-related behaviors were assessed. Immunofluorescence staining was performed to assess the expression of NE neurons in the LC, and Western blotting was used to determine the expression levels of α-2A-adrenergic receptors in the spinal cord nucleus caudalis (Sp5C). The authors concluded that the analgesic effect of taVNS is associated with the activation of the NE LC-28 system and the inhibition of Sp5C reduction in mice with migraine.
An interesting manuscript, and a very important and much needed topic - I have no objections to the concept, but there are a few questions and comments that could improve the manuscript.
Question regarding the mechanism of migraine and possible therapeutic targets: The Activity Regulated Cytoskeleton Associated Protein (ARC) plays a role in synaptic plasticity and is involved in neurological disorders such as migraine. Research suggests that targeting Arc may be a potential treatment for migraine. I would suggest using new publications on ARC, including (doi: 10.3389/fneur.2023.1201104.) and considering whether ARC can be related to the mechanisms that the authors describe.
>> A glass electrode filled with paraffin was used to extract 500 nl of virus (rAAV-hSyn-GRAB-190 NE2m(3.1)-WPRE-pA, BrainVT A (Wuhan) Co., Ltd, China), which was injected at a rate of 40 nl/min after proper positioning.
I'm not sure I understand this sentence correctly. How was the virus extracted and from where? I think it would be better to rewrite this sentence.
The authors rightly point out that SNRIs have indeed proven efficacy and may be the most effective treatment for patients with depression and migraine. This is a very broad and interesting question, I would suggest discussing it more widely? To what extent do migraine and depression share common biological mechanisms? Could these mechanisms have a local character, leading to epileptic seizures in focal epilepsy?
There is a vast literature on this topic, for example (doi: 10.1097/j.pain.0000000000001421), (doi: 10.1177/03331024241230466)
The manuscript is impressive, I will be happy to recommend the manuscript for publication after corrections have been made.
Author Response
Comment 1
An interesting manuscript, and a very important and much needed topic - I have no objections to the concept, but there are a few questions and comments that could improve the manuscript.
Question regarding the mechanism of migraine and possible therapeutic targets: The Activity Regulated Cytoskeleton Associated Protein (ARC) plays a role in synaptic plasticity and is involved in neurological disorders such as migraine. Research suggests that targeting Arc may be a potential treatment for migraine. I would suggest using new publications on ARC, including (doi: 10.3389/fneur.2023.1201104.) and considering whether ARC can be related to the mechanisms that the authors describe.
Response 1
Thank you for your suggestion. In this study, we did not specifically investigate the effect of taVNS on specific neurons of Sp5C. Your viewpoint provides us with a new perspective. In addition to exploring the mechanism by which taVNS alleviates migraine by regulating the LC-NE system, the active regulation of cytoskeleton-associated proteins (ARC) may be another molecular mechanism. As a key regulatory factor of synaptic plasticity, ARC plays a significant role in the formation of connections between neurons, especially in glutamatergic neurons [1]. Studies have shown that abnormal expression of ARC is closely related to neurological disorders including migraines. Therefore, ARC may become a potential target for taVNS in the treatment of migraine.
However, we did not observe the expression changes of ARC, but it cannot be ruled out that this molecular mechanism may be a target for taVNS in the treatment of migraine. It is hypothesized that taVNS not only regulate neural circuits through the LC-NE pathway, but also may affect ARC expression, further regulating synaptic plasticity and helping to restore normal neural function. Future research can further explore the regulatory effect of taVNS on ARC expression and evaluate its role in the LC-NE pathway to deepen our understanding of the molecular mechanism of taVNS in the treatment of migraine.
[1]Sibarov DA, Tsytsarev V, Volnova A, Vaganova AN, Alves J, Rojas L, Sanabria P, Ignashchenkova A, Savage ED, Inyushin M. Arc protein, a remnant of ancient retrovirus, forms virus-like particles, which are abundantly generated by neurons during epileptic seizures, and affects epileptic susceptibility in rodent models. Front Neurol. 2023 Jul 7;14:1201104. doi: 10.3389/fneur.2023.1201104IF: 2.8 Q2 . PMID: 37483450; PMCID: PMC10361770.
Comment 2
A glass electrode filled with paraffin was used to extract 500 nl of virus (rAAV-hSyn-GRAB-190 NE2m(3.1)-WPRE-pA, BrainVT A (Wuhan) Co., Ltd, China), which was injected at a rate of 40 nl/min after proper positioning.
I'm not sure I understand this sentence correctly. How was the virus extracted and from where? I think it would be better to rewrite this sentence.
Response 2
Thank you for your suggestion. The sentence has been revised to ”A microinjection syringe equipped with a glass electrode filled with paraffin was used to extract 500 nl of the virus (rAAV-hSyn-GRAB-190 NE2m(3.1)-WPRE-pA, BrainVTA (Wuhan) Co., Ltd, China) from the virus reagent tube. Then the virus was injected into Sp5C at a rate of 40 nl/min.”
Comment 3
The authors rightly point out that SNRIs have indeed proven efficacy and may be the most effective treatment for patients with depression and migraine. This is a very broad and interesting question, I would suggest discussing it more widely? To what extent do migraine and depression share common biological mechanisms? Could these mechanisms have a local character, leading to epileptic seizures in focal epilepsy?
There is a vast literature on this topic, for example (doi: 10.1097/j.pain.0000000000001421), (doi: 10.1177/03331024241230466).
Response 3
Thank you for your suggestion. Indeed, depression and migraine share some similarities in their biological mechanisms, particularly in neurotransmitter regulation, such as serotonin, norepinephrine, and dopamine, which play crucial roles in mood regulation and pain processing. Serotonin norepinephrine reuptake inhibitors (SNRIs)
including venlafaxine and duloxetine have evidence for efficacy and may be the most effective treatments in patients with comorbid depression and migraine[1]. Our research showed that taVNS exert analgesic effects on migraine through the downward inhibitory pathway of NE system. It indicates that taVNS also shows great potential in pain and depression comorbidities. In our another unpublished study, it was shown that taVNS could treat comorbidities of pain and depression. This information has been added in red in the revised manuscript.
The idea of whether these common mechanisms have local characteristics and may lead to focal epilepsy is very interesting. Although both migraine and epilepsy involve abnormal neural excitability, it is not entirely clear whether the same mechanisms that cause migraine and depression directly trigger focal epileptic seizures. However, an increasing amount of evidence suggests that cortical dilation suppression (CSD) that occurs in migraine and epilepsy may be a link between the two diseases. The characteristic of CSD is a wave of neuronal depolarization, followed by the inhibition of neuronal activity. This phenomenon is associated with the onset of migraine auras and epileptic seizures[2][3]. However, Due to the complexity of its mechanism and it has no direct relation to this study, and thus this part is not included in our discussion. Nevertheless, this will help expand the scope of our research and further explore how these conditions are interrelated at the mechanism level.
[1]Burch R. Antidepressants for Preventive Treatment of Migraine.Curr Treat Options Neurol. 2019 Mar 21;21(4):18. doi: 10.1007/s11940-019-0557-2.
[2]Liu TT, Chen SP, Wang SJ, Yen JC. Vagus nerve stimulation inhibits cortical spreading depression via glutamate-dependent TrkB activation mechanism in the nucleus tractus solitarius. Cephalalgia. 2024 Feb;44(2):3331024241230466. doi: 10.1177/03331024241230466. PMID: 38329067.
[3]Kramer DR, Fujii T, Ohiorhenuan I, Liu CY. Interplay between Cortical Spreading Depolarization and Seizures. Stereotact Funct Neurosurg. 2017;95(1):1-5. doi: 10.1159/000452841. Epub 2017 Jan 14. PMID: 28088802.
Round 2
Reviewer 1 Report
Comments and Suggestions for Authors
Dear author,
After careful revision, the manuscript was revised successfully and can proceed to publication.
Reviewer 2 Report
Comments and Suggestions for Authors
Well done
Reviewer 3 Report
Comments and Suggestions for Authors
In raw images, the molecular weight ladder bands appear consistently narrower than the sample lanes. Can you explain this discrepancy? Why do the marker ladder bands show unexpected uniformity in size and shape across different loadings, where I would anticipate minor variations? Why does the piece of β-actin 3 blots appear to be cut and placed on top of the whole membrane in the raw image?
Response to reviewer: Authors have re-run Western blots to show no fabrication of data.
Round 3
Reviewer 3 Report
Comments and Suggestions for Authors
Third review. No more comments.